# Computing Resource Allocation Scheme for DAG-Based IOTA Nodes

**DOI:** 10.3390/s21144703

**Published:** 2021-07-09

**Authors:** Houssein Hellani, Layth Sliman, Abed Ellatif Samhat, Ernesto Exposito

**Affiliations:** 1LIUPPA, Université de Pau et des Pays de l’Adour, E2S UPPA, Liuppa, 64600 Anglet, France; ernesto.exposito@univ-pau.fr; 2EFREI Engineering School-Paris, AllianSTIC, 94800 Villejuif, France; 3Faculty of Engineering-CRSI, Lebanese University, Hadat 1003, Lebanon; samhat@ul.edu.lb

**Keywords:** IoT, DLT, IOTA, Tangle, resource allocation, load balancing

## Abstract

IOTA is a distributed ledger technology (DLT) platform proposed for the internet of things (IoT) systems in order to tackle the limitations of Blockchain in terms of latency, scalability, and transaction cost. The main concepts used in IOTA to reach this objective are a directed acyclic graph (DAG) based ledger, called Tangle, used instead of the chain of blocks, and a new validation mechanism that, instead of relying on the miners as it is the case in Blockchain, relies on participating nodes that cooperate to validate the new transactions. Due to the different IoT capabilities, IOTA classifies these devices into full and light nodes. The light nodes are nodes with low computing resources which seek full nodes’ help to validate and attach its transaction to the Tangle. The light nodes are manually connected to the full nodes by using the full node IP address or the IOTA client load balancer. This task distribution method overcharges the active full nodes and, thus, reduces the platform’s performance. In this paper, we introduce an efficient mechanism to distribute the tasks fairly among full nodes and hence achieve load balancing. To do so, we consider the task allocation between the nodes by introducing an enhanced resource allocation scheme based on the weight least connection algorithm (WLC). To assess its performance, we investigate and test different implementation scenarios. The results show an improved balancing of data traffic among full nodes based on their weights and number of active connections.

## 1. Introduction

Nowadays, IoT technology prevails in almost all the business sectors [1,2]. It turns into an indispensable technology partner that has considerable significance in the production process. Nevertheless, the heterogeneity and the ever-increasing amount of IoT devices require a proper management system to address the enormous captured information [3]. To this end, in current centralized systems, many limitations prohibit the applications from the utility of the enormous IoT data, which probably leads to important information loss. On the other hand, the distributed ledger technology (DLT) responds to the IoT requirements [4] in regards to the integration with various IoT types. DLT enables the use of peer-to-peer (P2P) solutions where data are stocked in an immutable, tamper-proof, transparent, and traceable system [5]. The decentralized applications running on top of DLT satisfy the IoT requirements that bring many additional values to the business process.

Blockchain is the first DLT platform to run the full decentralized cryptocurrency, which falls under Blockchain version one. Blockchain evolves into two additional versions. Version 2 enriches the technology with the smart contract and the automation of the whole decentralized process. Version 3 introduces the decentralized applications (dApp) running on the participated nodes where all its data transactions are stored in that immutable ledger. When it comes to integrating Blockchain with the massive IoT devices, many limitations hinder this progression [6]. Blockchains, in general, are computationally expensive and involve high bandwidth overhead and delays, which are not suitable for most IoT devices [7]. Currently, the researchers direct their focus towards facilitating Blockchain integration with the industry through the fourth upcoming version. Besides, another DLT platform named IOTA is introduced to tackle the proliferation of IoT devices. IOTA [8,9] is a DLT solution primarily designed to respond to the massive IoT requirements. It is based on directed acyclic graph (DAG) with a specific ledger structure named “Tangle”. DAG is advisable for the IoT environment since it can simultaneously address the incoming transactions (Txs) [10]. These two DLTs are distinguished by their different data structure. Blockchain is based on a series of blocks that includes transactions, while IOTA is based on a graph where transactions are validated instantly [11].

IOTA categorizes its network participants into full and light nodes. The full node stores the IOTA ledger (Tangle), computes Txs and attaches them to the Tangle directly. The light node is a device with low resources that requires utilizing full node resources to validate and attach its Tx to the Tangle. Hence, an efficient mechanism within the IOTA platform to allocate full nodes’ resources is required. To achieve this goal, IOTA introduces the adaptive proof of work (PoW) [12] to push full nodes issuing Txs fairly that will be applied in the last release [13]. The adaptive PoW is a new approach that aims to assign different PoW difficulties to fit with the light devices and permit them to participate in the network. This method follows a solution based on reducing the computing difficulty to a certain limit [14] for low resources devices. However, the following points motivate us to propose a new resource allocation mechanism:▪A light node is not directly connected to the Tangle; instead, it is connected to a full node. Light nodes consume the full node’s resources arbitrarily to validate and attach their Txs to the tangle [5]. The light nodes connections are not distributed fairly between full nodes. It happens that a full node has a high number of linked connections than others. Thus, it performs huge computing tasks while other full nodes are in idle state, leading to a performance issue during the peak time;▪The established connection between light and full node is unstable since the latter is not replicated, and it is not guaranteed to be online all the time. This type of connection is considered as a single point of failure.

Thus, an efficient mechanism is required to allocate the full nodes’ resources with load balancing. This paper proposes a resource allocation scheme to fairly redistribute the decentralized computing loads between the IOTA full nodes. The target is to balance the computing tasks among all full nodes. This can be achieved by the collaboration between the nodes to maintain the maximum system performance.

The main contributions of this paper are:
Shed light on IOTA as an alternative DLT platform satisfies the massive IoT devices;Highlight the interconnection mechanism of different IOTA nodes and the resource allocation concern in the current situation;Propose a load balance scheme to redistribute the computing tasks fairly based on the different node capabilities.

The rest of this paper runs as follows: Section 2 overviews the Blockchain and elucidates the DAG technology to provide the required background information regarding IOTA platform. We first present the Tangle components, and we depict an overview of the IOTA and DAG structure in addition to the recent IOTA updates. Section 3 proposes a resource allocation scheme where light nodes use the full nodes’ resources to achieve their tasks with load balancing. The validation of the proposed resource allocation is done in Section 4, and we conclude the paper in Section 5.

## 2. Background and Distributed Ledger Technologies

Before delving into the approach, we provide a clear overview of the used or employed technology and an outline of the proposal features.

### 2.1. Blockchain Overview

Blockchain is a distributed ledger that records all the transactions that have occurred among the participants of its network [5]. Therefore, the majority of the participants should validate the transactions to be involved in the ledger. Once a transaction is recorded, there is no way to alter or remove it. The validation is the core process of Blockchain which depends on the utilized consensus algorithm. Many consensus algorithms can run within the decentralized network, such as proof of work (POW), proof of stake (POS), proof of authority (POA), etc. Each consensus type depends on the type of Blockchain. The three main Blockchain types are public, private, and consortium. Public Blockchain is a permissionless distributed ledger where any node can join the network and perform transactions without authenticating with a third party. The private Blockchain is a controlled distributed ledger, where the owners manage the validation process. Finally, the consortium Blockchain is a permissioned Blockchain, composed of multi-organizations, such as bank branches.

Figure 1 depicts the Blockchain data structure that consists of an infinite chain of blocks started with a block known as ‘Genesis’. All the other block consists of a header and a body. The latter contains the validated transactions, where the header contains various fields responsible for maintaining the chain. The header fields are mainly composed of block version, hash of previous block header, timestamp, and Merkle tree hash representing the hash value of all the transactions in the block.

### 2.2. IOTA Infrastructure

The IOTA Foundation has developed the Tangle as an alternative to the Blockchain to tackle most of the current blockchain drawbacks [8]. There are two main motives behind IOTA conception: i. the necessity of a scalable ledger because of the massive transactions sent by large IoT devices and ii. the micro-payments of these devices. IOTA is based on DAG [15], where the transaction is the only element of IOTA. Precisely, there is neither a block nor a chain. DAG is a mathematical graph approach, which consists of an edge and a vertex. The edge is a unidirectional vector between two vertices with no loop back to the initial vertex.

As shown in Figure 2, a Tx represents the vertex of DAG, which is charged by the validation of two Txs as a condition to be issued. Thus, to issue a transaction, the intended node should establish two direct connections to two different Txs. Connecting these Txs in this binary form and the upcoming arrival Txs attached to the network determine the shape of the Tangle, as shown on the right side of Figure 2. The genesis is the first Tx of the Tangle where tokens are created.

Physically, each participating node communicates with its neighbor nodes to replicate their data, where all the nodes share the same ledger. The new Tx ,which is recently attached to the Tangle and is not validated yet by any node, is called “tip”, and is highly recommended to be selected by other nodes. A node that aims to issue Tx, selects two Txs from its up-to-date ledger to be validated; thereby, it will approve or decline them based on the ledger content. The lazy node is defined as the node that aims to pick approved Txs rather than tips. Selecting Txs, which are already validated, will leave behind many tips orphaned and impact the overall system performance. Therefore, it is recommended to eliminate the lazy nodes in the first step, or in other terms, encourage nodes to select tips rather than non-tips Txs.

#### 2.2.1. Coordinator

The Tangle’s security is primarily based on the high frequency of Txs that allow a considerable amount of nodes to participate in the Tx validation. The Tangle turns to be more secure when more Txs are issued. The more Txs, the more secure is the Tangle. Currently, the Tangle network’s size cannot achieve robust protection against several attacks where the success of double-spending is possible. Accordingly, the coordinator [15] is implemented temporarily as a trusted third party to check the Tangle health periodically. It is a centralized application operated by the IOTA foundation to protect the Tangle from different kinds of attacks and double-check the validation of the transactions. The coordinator issues a milestone transaction every two minutes to validate the Txs. All the approved Txs have an immediate confirmation confidence of 100%. Consequently, the second Tx of a double-spending will not be accepted, if any. Each transaction attached to the Tangle has its parameter values [15] that the coordinator uses to determine its path, and thereby, the honest Txs have a higher probability of being validated, while the lazy nodes are punished.

#### 2.2.2. IOTA Address

Each node’s user has their wallet composed of a “seed” address that generates an unlimited number of public or private addresses. The newly generated addresses are retrieved from the combination of a seed plus address indexes. The address index is a positive integer starting from “0”. Consequently, the “seed” is a random combination of 81 characters (letters A–Z) and the number nine that represents the secret account key, which should not be disclosed by anyone except the wallet owner. When a user publishes a Tx to the network, Tx will be signed based on Winternitz-One-Time-Signature-Scheme [16], and a part of their private key address is disclosed to the public.

#### 2.2.3. Creation Transaction Mechanism

For a successful Tx issuing, a node is responsible for applying three steps sequentially: creating a bundle, selecting and validating two tips, and performing PoW:Create a bundle: whenever a node wants to add a Tx to the network, it should create a bundle of Txs called sub-Txs. A normal Tx is a bundle of four sub-Txs that are indexed from 0 to 3. Index 0 is the recipient’s address “output” of the external wallet with the amount to be sent. Index 1 is the sum of all the amounts inside the sender wallet called “inputs” and has half of the sender’s signature. Index 2 represents the second half of the sender’s signature. Index 3 is the remainder “output” that must return to the sender’s wallet, which is the output minus the input. Accordingly, successful bundle results in equal number of Txs in both output and input. The bundle is atomic so that either all its Txs are accepted or none of them.Select tips: Before attaching a Tx (bundle of Txs) to the Tangle, the node should select the two newest tips and approve them.Building a robust “select tip algorithm” is mandatory in a DAG-based decentralized environment. The motivations behind designing such an algorithm are based on pushing nodes to select the unapproved Txs (tips) and checking the conflict of transactions, double spending, and falsifying. There are three main algorithms that the coordinator can use to trigger the node to select tips [17]: unweighted random walk, weighted random walk, and Markov Chain Monte Carlo (MCMC).Proof of work (PoW): Performing PoW is the node’s last task before issuing its transaction. Once the bundle is created, signed, and tips are attached to the bundle, the node performs PoW for each Tx of the bundle. The PoW is a sophisticated mathematical approach represented by the node’s computational effort to achieve a predefined minimum weight magnitude (MWM) of the hash function Curl [16]. An MWM is the number of zeros included in a nonce to be accepted. During the execution of PoW, a nonce is found by combining a specific counter with Tx data that fit with the MWM. The PoW process is hard to achieve, but it is easy to verify the answer. Thus, the PoW validation will be the node task that signalizes this tip, and so on. Once PoW is performed, the bundle is attached to the Tangle as a new tip and broadcasted to the whole network to be validated by some node(s) later on.

### 2.3. IOTA Updates

Since the very beginning, the IOTA foundation recognizes the necessity of eliminating the coordinator to be a fully decentralized system once the network becomes big enough. The coordinator plays significant roles in security and achieving consensus, so to shut it down, the core network will witness major enhancements over Tangle components. Thereby, they are working on fixing several drawbacks in the next release by introducing “the Coordicide” [13]. The coordicide includes the below new features that reshape the next IOTA structure:Autopeering: Currently, the process that allows nodes to join the network is applied manually, but it probably subjects the nodes to various attacks, such as the eclipse attack. Where the adversary can control the entire neighbor’s node. The autopeering mechanism is required to facilitate the neighboring operations and hinder attacker activities from targeting specific nodes. Autopeering consists of “peer discovery” and “neighbor selection” mechanisms. Peer discovery uses the authentication ping-pong protocol that allows every node to impart or perceive other network participants.Voting and Consensus: Tangle can comprise conflicting Txs due to the network propagation delay. Thus, it is required to reach a consensus on those conflicting transactions, which are currently applied by the tip selection algorithm. However, this algorithm is considered slow in solving the conflict since it uses random walking bias via honest nodes that leave conflicting branches behind. In addition, the Txs that select the wrong branch will be orphaned and reattached to the large number of Txs of the proper branch. Therefore, a consensus mechanism called “Shimmer” is introduced in the new release of IOTA. Hence, nodes query other nodes about their current opinion of the ledger and adjust their opinion based on the proportion of other opinions. Two voting mechanisms, fast probabilistic consensus (FPC) [18] and cellular automata (CA) [19] are used to allow nodes to communicate and decide on Txs status. There is a possibility of requiring a combination of both mechanisms to add flexibility to the voting process.Tip selection: Represents a crucial part of the IOTA network that pushes nodes to verify the Tangle Txs. Currently, the biased random walk used by the coordinator has computational drawbacks, as it adds complexity over the orphaned Txs and obligates nodes to reattach them later on. The new consensus mechanism is independent of the tip selection algorithms (TSA), so the current TSA algorithms can be enhanced to select faster tips and incentivize non-lazy nodes. Furthermore, the limitation behind the biased random walk behavior is improved by pushing the node to select from non-lazy nodes only while lazy nodes still have a chance to be promoted and approved if their issuer intends to. There is no direct intervention in the selection process, and at the same time, the selection becomes faster;Adaptive PoW: IOTA proposed this new algorithm [13] to allow devices with low computing resources to be involved in the attaching Txs’ process to the Tangle. Additionally, it seeks to limit the devices with high resources from attaching an infinite number of Txs. IOTA defines new parameters on each node. The basic difficulty represents the threshold difficulty level that fits with any small device capacity. The adaptive rate is calculated based on mana owned by the node [13] and the number of Txs issued by this node within a time *w*. Each node’s new difficulty is equal to the basic difficulty plus the adaptive rate multiplied by the number of issued Txs within a time interval *w*. Thus, the more a node issues Txs, the more the difficulty increases, and the allowed number of Txs is adjusted. On the other side, this algorithm empowers the low-resource devices to issue Txs with a minimal degree of difficulty;Global node identities: in the new coordicide architecture, each node in the network has its identity that must be well protected. The identity is based on a new common public-key cryptography created by the IOTA foundation to sign Txs and link it to the issuing node in a tamper-proof way. Additionally, the issuing node adds its public key to every signed Tx. On the other side, introducing identities leads to a Sybil attack [20]. By introducing “mana”, this kind of attack is mitigated. “Mana” is a reputation value that is equivalent to the total funds transferred within the transaction. In that way, the more mana, the more contributions in the network and vice versa.

## 3. Resource Allocation Proposal

As mentioned previously, a decentralized network consists of various IoT devices, PCs, and servers distinguished by their hardware resources, including computing power and storage capacities. The IOTA decentralized performance depends on these resources to run the platform. Therefore, IOTA categorizes the devices participating in the network into full and light nodes. The full node stores the IOTA ledger (Tangle), computes Txs and attaches them to the Tangle directly. The light node is a device with low resources linked indirectly to the Tangle through any active full node. The light nodes randomly try to select their full nodes. As a result, many light nodes connect to a few full nodes while other full nodes are almost idle, as illustrated in Figure 3. Noticeably, this random connection does not consider the capacity of the selected full nodes. Thus, it affects the whole system performance and encourages us to propose a new mechanism to distribute the loads equally among full nodes based on their different resources. The IoT nodes are categorized into three main types:Full node: is similar to the full node categorized by IOTA. Such nodes are essential in the P2P system and are characterized by the full ledger size and high computing power. Full nodes are the only components of the IOTA network to attach the Txs to the Tangle;Light node: also similar to the light nodes categorized by the IOTA; it is the node that has computing capacity much less than the full nodes and higher than the zero nodes (defined below). The light node can create and sign Tx, but it does not store the ledger or attach its Tx to the Tangle directly;Zero node: the node that does not share its resource with any node and requests assistance from other nodes to attach its Tx(s) to the Tangle.Zero nodes are divided into two categories: permanent and temporary. The permanent zero nodes represent the weak IoT devices that cannot perform computing effort or store ledger information. This type of zero nodes does not participate in the Tangle network directly. However, they are attached to one of the active nodes. They are similar to the lightweight node in the current IOTA classification and assigned to nearly similar tasks. The second category is a temporary zero node that is one of either full nodes or light nodes which stops sharing their resources with other nodes and decides to request assistance from other nodes according to our algorithm rules. An active full node is turned into a temporary zero node in the below cases:
✠High traffic: an active node with a high number of Txs that bypasses a predefined limit. It forcibly turns into a temporary zero node and redistributes the incoming Txs to all other nodes based on the proposed resource allocation algorithm;✠Offline status: in case of maintenance, loss of connection, or any other hardware failure, the node will be suppressed from all the neighbor lists. However, it can generate offline Txs in some cases;✠Owner decision: The user can manually turn off the share node activities.

Table 1 shows our classification versus the current IOTA classification in terms of the main network tasks. The new classification adds flexibility to the network by empowering full nodes to manage the computing requests and share their free resources with other nodes. For example, in a cluster of different neighbor nodes, the full nodes cooperate according to a new resource allocation algorithm to perform the Tx requests from the light, permanent, and temporary zero nodes.

### 3.1. Load Balancing Overview

Load balancing algorithms are broadly classified into dynamic and static categories [21]. The latter depends on the load at the time of node selection, whereas it is achieved by providing preliminary information about the system. This approach does not consider the system’s current state while making allocation decisions, and, therefore, it weakens the overall system performance [22]. Obviously, the static approach is unsuitable for DLT systems since the participant nodes frequently enter and exit the network without prior notifications. Dynamic load balancing algorithms perform load distribution at runtime [23] and monitor any alteration of the system workload to redistribute the tasks based on the current state of the whole system. Usually, dynamic load balancing is considered a central solution that acts as a proxy between end-users and servers.

In the literature, there are several types of dynamic load balancers. For example, round-robin scheduling [24] directs the application requests from the network to nodes in a round-robin manner. It is widely used and easy to implement. However, it considers all nodes are equals in terms of resources and number of connections and unsuitable for our case. Weighted round-robin is an improved version of round-robin where a weighted coefficient is assigned to the node according to its capacity. Nevertheless, such weight estimation does not take into account the number of current connections. Another algorithm based on the number of active connections is the least-connection scheduling algorithm that assigns the received requests from the network to the node with the least number of established connections. In our case, the limitation of this algorithm refers to neglecting the nodes’ resources. The weighted least-connection scheduling is a developed version of the least-connection scheduling that considers both the capability of the node and the number of current connections to prevent overloading and achieve load balancing. Thus, our proposed resource allocation scheme to distribute computing loads between the IOTA full nodes is based on WLC.

### 3.2. WLC Algorithm

WLC algorithm considers a superset of active connections on each node and its assigned weight based on the node processing capabilities [25]. This algorithm is used in a centralized environment, such as web server, SQL Server, etc., to load balancing the incoming traffic from the clients’ side. In our use case, the contest is to adapt WLC to a decentralized system where no one node can dominate the load balancing role and task distribution.

The network nodes share their nodes and connection number with their neighbors to achieve the WLC load balancing in such decentralized environment. Nevertheless, each neighbor list is up to nine nodes maximum [8,26]. This is a good point to limit the negative impact of the overhead due to the communications.

Each node of the system uses its hardware resources to retrieve the initial weight and uses the number of connections and the sum of its neighbors’ connections parameters. Hence, each node in the network, considered as a load balancer, will be active whenever it is assigned to an external task. Therefore, at the time of request arrival, the node runs the WLC algorithm against its weight and other nodes’ resources to assign the request.

Once the request is assigned, the remaining weight value of the assigned node decreases to be checked with the next incoming request, and so on. In this way, all the nodes contribute to balancing the loads among them and fulfill the scalability and performance enhancements without a central load balancer.

Supposing the neighbor set of n nodes in a network is:
I={1,2,3…,n}

Node *i* has $W_{i}$ Weight and $C_{i}$ number of connections. The sum of current connection numbers is:$$C_{SUM}=\sum_{i\in I}C_{i}$$

The incoming network connection will be directed to the node j, in which$$\frac{C_{j}/C_{SUM}}{W_{j}}=min_{i\in I}\left\{\frac{C_{i}/C_{SUM}}{W_{i}}\right\}$$

Since the $C_{SUM}$ is a constant in this lookup, there is no need to divide $C_{i}$ by the $C_{SUM}$ value. Thus, it can be reduced to be as below:(1)$$\frac{C_{j}}{W_{j}}=min_{i\in I}\left\{\frac{C_{i}}{W_{i}}\right\}$$

The weight of a node is equivalent to its computing resources and can be captured as a digit number. Let $V_{i}$ be the processing speed of the CPU of the node i (in MHz) and $C_{V_{i}}$ be the consumed capacity of the CPU of the node *i*. Thus, $Vr_{i}$ the idle processing ratio of the node i CPU is given by: $Vr_{i}=1-(C_{V_{i}}/V_{i}$ ).

Let $M_{i}$ be the Memory size of the node *i* (in MB) and $C_{M_{i}}$ be the consumed part of the memory of the node *i*. Thus $Mr_{i}$ the free memory ratio of the node i: $Mr_{i}=1-(C_{M_{i}}/M_{i}$).

The weight of node *i* is given as follows:(2)$$W_{i}=\left(\alpha \times Vr_{i}+\left(1-\alpha \right)\times Mr_{i}\right)\times 100$$

This weight $W_{i}$ is a load indicator between 0 and 100. Higher value of the weight reflects that the node is able to accept new request. 0<α<1 is a weight coefficient of the CPU idle ratio and (1−α) is a weight coefficient of the free memory ratio.

The component diagram of Figure 4 depicts the behavior of a new node that aims to join the network and initiate a new Tx. The node could be zero, light, or full. The network system filters out the device with weak resources, nominates it as a permanent zero node, and establishes a direct connection with one of the active full nodes based on the WLC algorithm. The zero node has no WLC algorithm running locally; instead, it triggers the full node to attach its Tx to the Tangle. The non-weak devices are either full or light nodes, where the full ones join the main Tangle network. The light node can possess the WLC algorithm and run it to select a suitable node. Then, the full node that receives the request runs the WLC to double-check that it is the best node in the group. Otherwise, the request will be directed to the best node. In the case of the active full node, the WLC permits to assign the Tx to the node itself if its workloads are below certain limits. Otherwise, the active node is considered fully charged and will not be considered in the current selection. It also turns into a temporary zero node for a while, and its Tx will be assigned to another available node according to the WLC rules. As per the peer-to-peer concept, all the nodes are free to leave and rejoin the network at any time. Therefore, the active node can turn into a temporary zero node for any reason such as maintenance mode, network disconnection, etc. Moreover, the selected node by the WLC algorithm can accept or refuse the computing request. Once a node accepts the task, it increases its connections by one and publishes the updated load balancer parameter to be available to the next node request and accessible by all the neighbors’ list members. This mechanism of a node, while initiating Txs, is repeated instantly for each node within its neighbors’ list to achieve the load balance of the computing tasks all over the neighbors’ lists of the DLT system.

## 4. Experiments and Results

The WLC implementation can be performed with different scenarios. It can be installed either on the light node, on the full node, or both. We start by testing the WLC algorithm on the light node against a few Tangle nodes. The WLC is implemented in a private Tangle network consisting of several virtual machines that act as full nodes with different specs, and one virtual machine works as a light node. All the nodes are connected to the same network. The light node is represented by IOTA wallet software that can create and sign Tx only. Therefore, it should be connected to a full node to perform PoW tasks and attach its Tx to the Tangle. In this scenario, the light node runs the WLC algorithm of [25] to connect to one of the nodes in the list. Thus, load balance is achieved based on the WLC decision. In addition to the experiment done above, the WLC algorithm is evaluated on the full nodes using a simulation environment through a Java-Based tool, demonstrating the effectiveness of the load balancing algorithm. α is the coefficient rate to determine the CPU/Memory resources importance of a node. α has a minor impact on the proposal result since the weight reflects the node resource. For example, setting α to 0.25 means that when calculating the weight of a node, we give less importance to the CPU rate than the memory, and so on. Finding optimal α value is out of the paper’s scope. In our evaluation, we set α = 0.5 (CPU and memory are equal), and we distinguish between different scenarios.

### 4.1. Implementation: WLC in a Private Tangle

In this experiment, the proposed algorithm is embedded within the light node, as shown in Figure 5a, that runs against the full nodes to select the best available node. In this scenario, the light node is the only part that determines its connection. The light node runs the directly involved WLC algorithm within the Tangle. Inspired by IOTA client load balancer [27], we create a node js application that replaces the random selection method (RandomWalkStrategy.js) with the WLC algorithm. We use the IOTA libraries to build the load balance algorithm that can be installed on light and full nodes. The libraries include @iota/core, iota/client-load-balancer, iota/converter. Besides, we use the “node-os-utils” library to determine the load information of each full node. The experiment’s result that is limited to a few nodes within a private Tangle network shows the tasks are distributed based on the highest weight and least connection of the WLC algorithm.

### 4.2. Simulation: Decentralized WLC

To build a network while increasing the Tangle nodes, we use a simulation environment based on the Java compiler to simulate the WLC behavior within a decentralized environment. We first build the datacenter class to store all the network components. The simulation code is available on GitHub repository [28]. The class node generates nodes with random CPU values ranging from 1 KHZ to 6 GHZ and random RAM ranging from 500 MB to 16 GB. Furthermore, a JSON file is introduced to manually determine the values of the resources. This file permits us to create nodes either with similar or different resources to test different scenarios, as shown in the next section. Each connection consumes a fixed amount of resources and deducts the same value from any assigned node. Accordingly, the weight of that node is updated according to Equation (Equation 2). We set the connection request parameters to 10 MHZ as CPU and 50 MB as RAM in all the below tests. The performance indicators resulting from the simulations are: the weight of the node that reflects the available resource of the node and the ratio indicator $C_{i}/W_{i}$ that reflects the load of the node.

#### 4.2.1. Nodes with Similar Resources

In this scenario, we validate the implemented WLC algorithm and consider a network of 16 nodes where all of which have same CPU and RAM resources as shown in Figure 6a with 1000 Mhz for the CPU and 4000 MB for the RAM. The initial network is built upon one default connection assigned to all nodes in the first place to distinguish the different initial resource weights. Without losing generality, this test assumes that all the incoming connections require the same resources in terms of CPU and RAM. We activate the WLC algorithm for the nodes of different incoming requests ranging from 50 to 1000 and record the snapshots output. Before any connection request, the maximum initial weight of each node, which represents their resource consumption and maximum capacity ratio, is close to 100. This number reflects that the nodes are still in an idle state. After being assigned to connection requests, the nodes’ resources decrease, and thereby, their weights decrease relatively, i.e., the available resources of the node decrease. As illustrated in Figure 6b, connections are distributed evenly over the nodes, and all nodes’ remaining weights decrease in a similar way because they all have similar capacities. Figure 6c highlights the weights of the nodes when the number of connections is between 50 and 65. Thus, the WLC algorithm distributes the tasks orderly and fairly.

#### 4.2.2. Nodes with Different Resources

Afterward, the second test distinguishes 16 nodes with variable weights as depicted in Figure 7a. We also assume that all the incoming connections require the same resources in terms of CPU and RAM. The target of this test is to generate the maximum traffic and monitor the WLC behaviors in terms of the remaining node resources. Accordingly, the number of connections is progressively increased, as illustrated in Figure 7b. For further clarification, Figure 7c details 25 successive incoming connections to these different nodes and shows the WLC behavior precisely. It is noted that with every new connection, the node with high weight and least connections is selected.

As the nodes are with different resources, we also plot the indicator $C_{i}/W_{i}$ in Figure 8a to illustrate loads of the different nodes within the system. Based on the WLC algorithm, the node with the minimum load is selected when receiving a request. One can see that the values of this indicator for the different nodes are close to each other. The result shows that the traffic is distributed equally among the 16 nodes based on their remaining weights and previous active connections. With the traffic increase, all nodes’ remaining weights decrease proportionally, demonstrating the elimination of the node overloading aspect. Figure 8b details the WLC behavior of the different nodes with connections ranging from 200 and 225. This test’s microscopic view is another proof of the nodes’ response to the WLC distributed load balancer.

### 4.3. Extension to a Tangle with Multiple Networks

In this case, the Tangle is composed of multiple networks each node has its own neighbor list based on the node distribution across their locations and their network addresses. Figure 9 depicts an example of a few full nodes distributed in three network clusters.

Node *i* has a neighbor list that contains all nodes so that all these nodes have node *i* as a common neighbor in their lists. Node *j* belongs to two neighbor lists, so its visibility is limited to list 1 and 3 only. Thus, Node *j* can assign tasks or it can be assigned to tasks by these two lists’ nodes.

However, other nodes like s,o have different neighbors that have, in turn, different neighbor nodes. Such behavior represents the main characteristic of decentralization where the information propagates from one node to another following the Gossip protocol [29]. The difference in these lists is normal in a P2P network that contains enormous number of participants. In addition, the neighbor list on each node is periodically updated since participants enter and leave the network randomly. The peer-to-peer system can scale to millions of processes, each of which can join or leave whenever it pleases without seriously disrupting the system’s quality of service [30].

To best manage the massive workload within the huge number of networks while assuring the load balancing of the incoming traffic, we distinguish between different neighbor lists by a different sum of connections $C_{SUM}$. In this case, the sum of connection should not be reduced, as it is not the same for the whole network. Thus, the WLC is implemented in each network where $C_{SUM_{1}}$ is the sum of the connections in the network 1, $C_{SUM_{2}}$ is the sum of the connection in the network 2 and $C_{SUM_{3}}$ is the sum of the connections in the network 3. As *j* belongs to two neighbor lists, when running WLC, Node *j* uses Min($C_{SUM_{1}},C_{SUM_{3}}$). Additionally, Node *s* uses Min($C_{SUM_{2}},C_{SUM_{3}}$), Node *o* uses Min($C_{SUM_{1}},C_{SUM_{3}}$) and Node *i* uses Min($C_{SUM_{1}},C_{SUM_{2}},C_{SUM_{3}}$). We simulate the three networks where the 16 nodes are with similar resources to validate the ability of the proposed mechanism to load balance the incoming connections. The simulation generates 1000 connections towards the three networks. In the beginning, the nodes have the same weight, and each network has an equal number of nodes, so the $C_{SUM_{1}},C_{SUM_{2}},C_{SUM_{3}}$ are equal. Once a connection is assigned to a node, the node’s resources and weight are updated, and the $C_{SUM_{h}}$ of the network *h* where the node exists will increase.

Figure 10 shows the increasing number of connections per network. One can see that the connection loads are distributed efficiently among the multiple networks. For example, in the case of 1000 connections, the results show a load balancing between networks and nodes where each node is allocated to around 63 or 64 connections. Note that the total number of connections is greater than 1000 because the connections of the node *j* are counted in network 1 and also in network 3. Similar remarks for the nodes *o*, *s*, and *i* according to the corresponding networks.

## 5. Conclusions

In this paper we enhanced the distribution of tasks among different nodes of the IOTA platform. This is done by introducing a resource allocation scheme to handle the incoming workload transactions. Our proposal is based on the WLC scheduling load balancing algorithm which distributes the incoming external requests fairly among nodes based on their resource capacities (weights) and connection numbers. The proposed algorithm is tested in a private Tangle and through simulations. The results denote an efficient selection of the suitable node based on the available resources and active connections. In this work, we used α = 0.5. Future work will focus on finding the optimal value of alpha to determine the weight coefficients of both the CPU idle ratio and the free memory ratio.

## Figures and Tables

**Figure 1 sensors-21-04703-f001:**
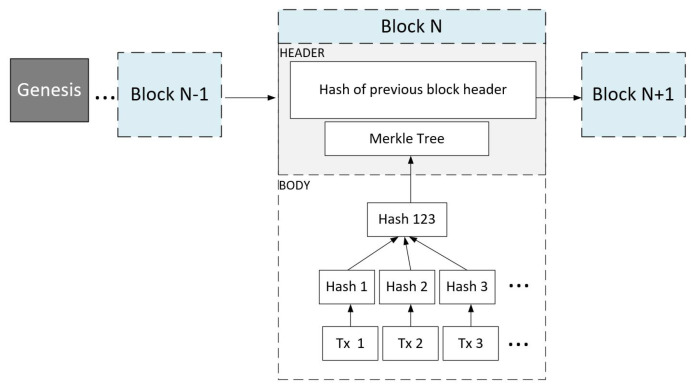
Blockchain Structure.

**Figure 2 sensors-21-04703-f002:**
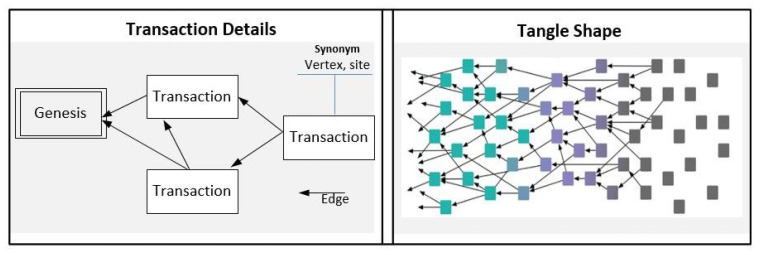
IOTA DAG Structure.

**Figure 3 sensors-21-04703-f003:**
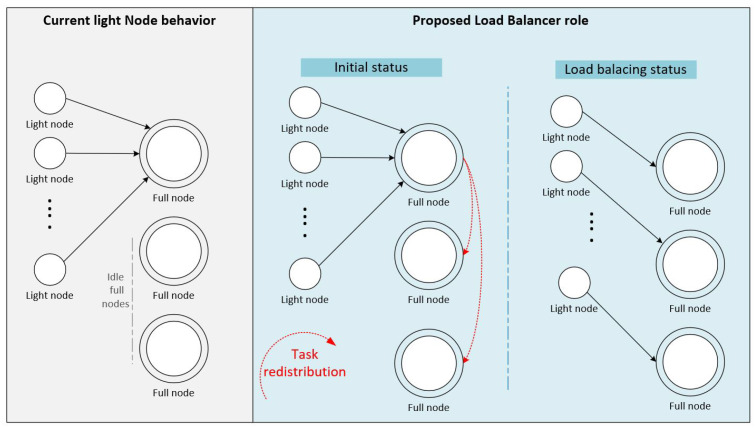
Load balancer role: each node should act as a load balancer to distribute the incoming random tasks fairly among full nodes.

**Figure 4 sensors-21-04703-f004:**
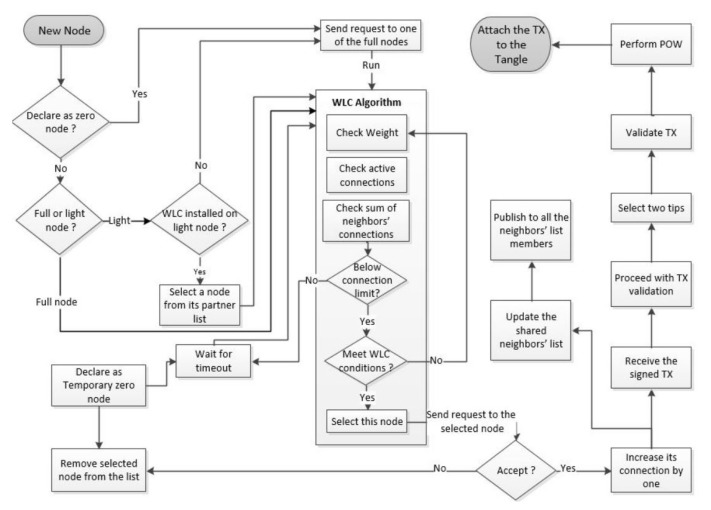
Flowchart diagram of the proposed solution.

**Figure 5 sensors-21-04703-f005:**
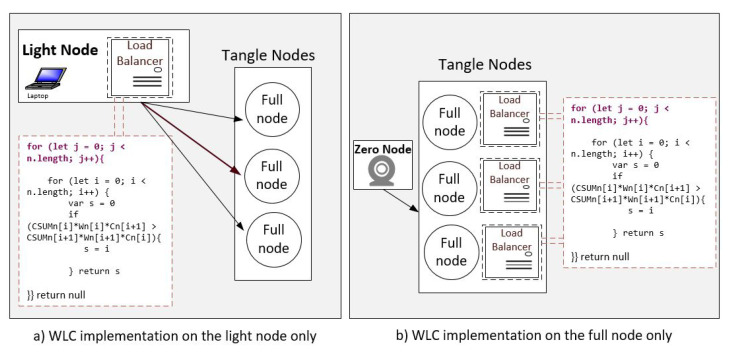
WLC implementation with zero node and light node scenarios. With zero node scenario, the WLC is running on the full node network only. With light node scenario, WLC is running on light node directly.

**Figure 6 sensors-21-04703-f006:**
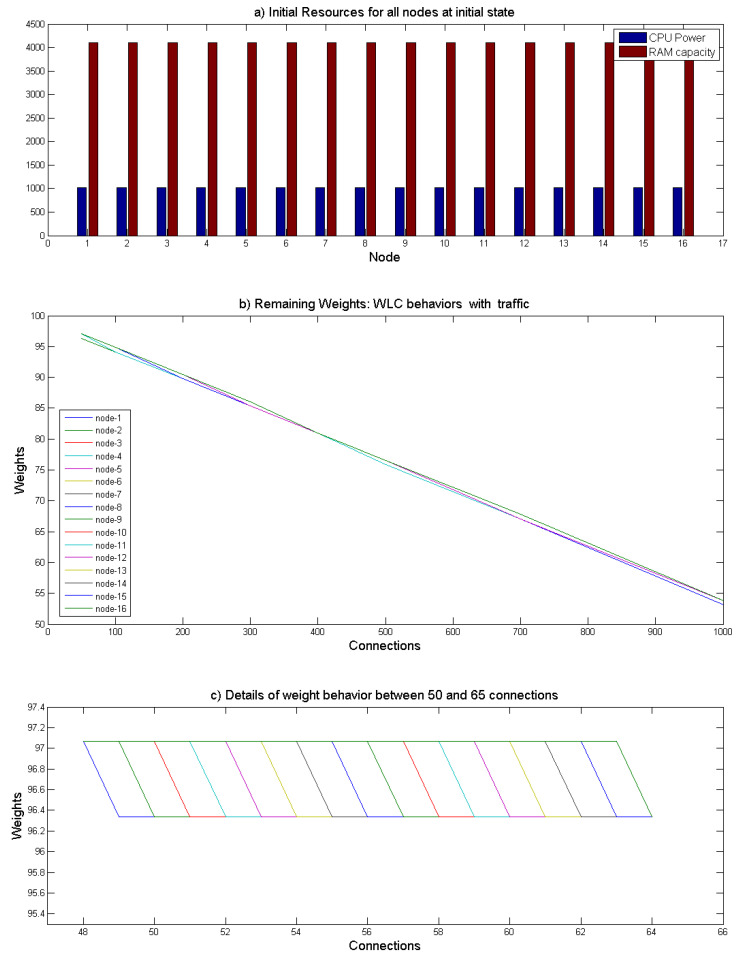
Validation of the WLC algorithm using 16 similar nodes. the WLC behavior demonstrates the validity of the WLC algorithm.

**Figure 7 sensors-21-04703-f007:**
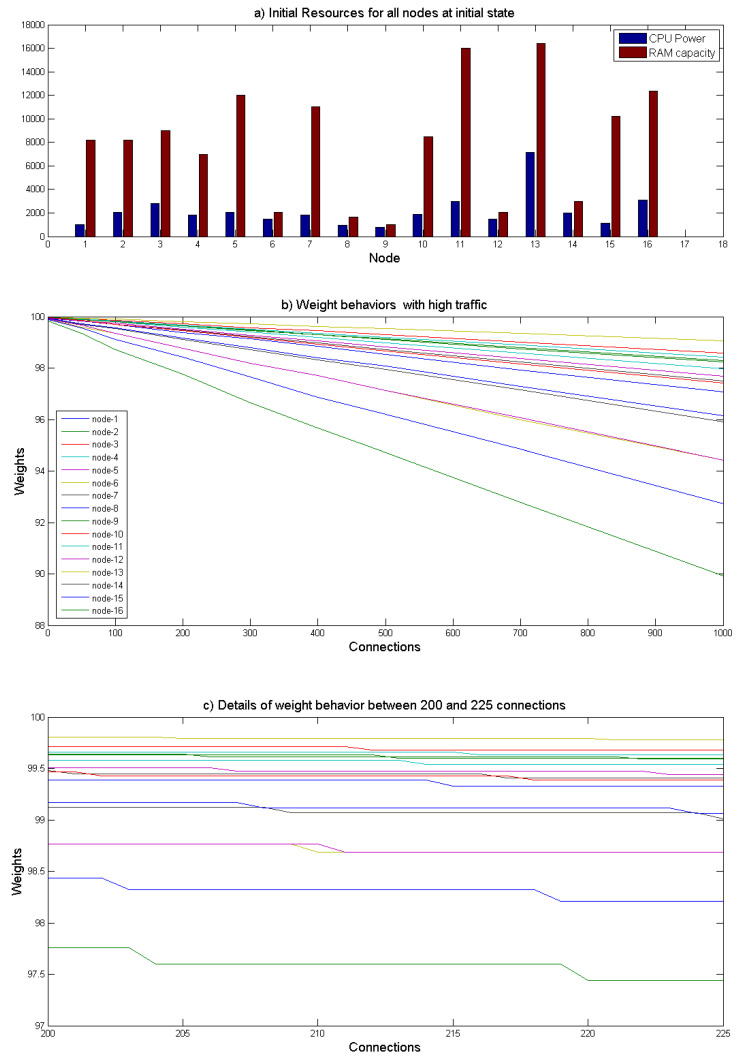
Nodes with different resources.

**Figure 8 sensors-21-04703-f008:**
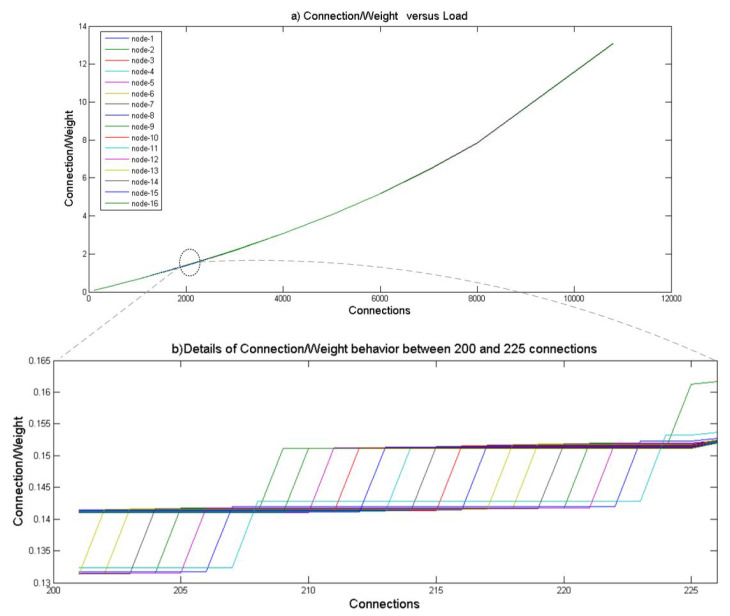
Nodes with different resources: $C_{i}/W_{i}$.

**Figure 9 sensors-21-04703-f009:**
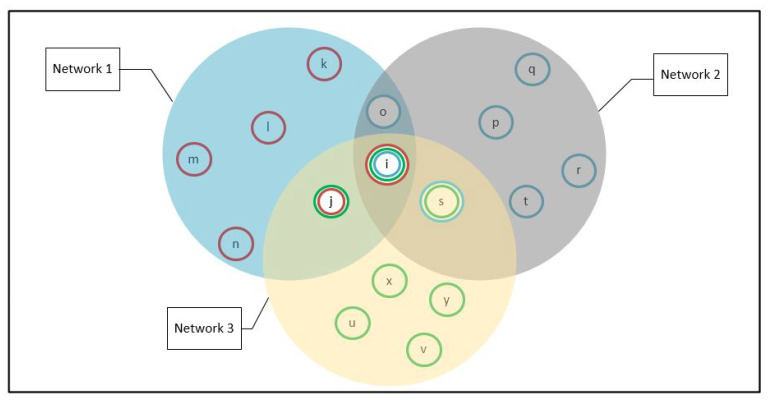
Neighbor lists of the adjacent nodes.

**Figure 10 sensors-21-04703-f010:**
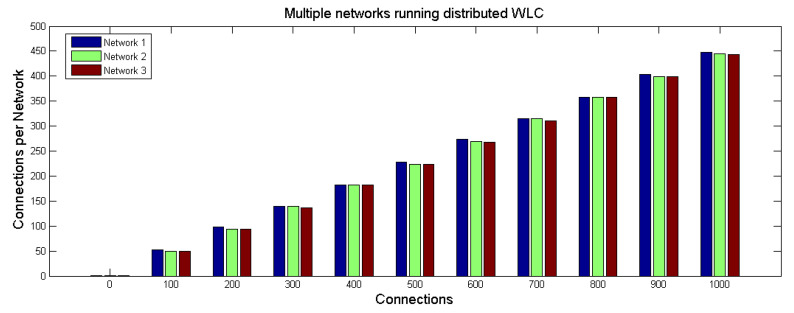
Multiple networks running distributed WLC.

**Table 1 sensors-21-04703-t001:** IOTA node capabilities.

Functions	IOTA Structure		Proposed Structure			
	Full Node	Light Node	Full Node	Light Node	Temporary Zero Node	Permanent Zero Node
Stores the Tangle	✓	✕	✓	✕	✓	✕
Communicate with neighbors	✓	✕	✓	✕	✓	✕
Bundle, create, sign tx	✓	✓	✓	✓	✓	✓
Tip selection	✓	✕	✓	✕	✓	✕
Validation	✓	✕	✓	✕	✓	✕
POW locally	✓	✓	✓	✕	✓	✕
Attach to Tangle	✓	✕	✓	✕	✓	✕
Receive Transaction request	✓	✕	✓	✕	✕	✕

## Data Availability

Not applicable.

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
