# Peer review of "Computing Resource Allocation Scheme for DAG-Based IOTA Nodes"

_sensors, 2021, doi:10.3390/s21144703_

Round 1

Reviewer 1 Report

The paper describes an allocation scheme for an IoT-oriented distributed ledger technology. The solution is based on the WLC (Weight Least Connection) algorithm. An experimental evaluation is provided.

The topic of distributed ledgers that suit the needs of data processing at the edge and the IoT/sensor network context is highly topical. In this case the proposed research is geared towards the graph-based tangle structure of IOTA.

The author demonstrate competence in their technical work. The technical part appears to be technically sound.

I found it hard to understand the objectives of the paper. It is not clearly enough formulated that the aim is distributed management (load allocation/balancing) of IOTA itself and not the applications using IOTA possibly partly for this purpose.
Without blockchain/distributed ledger background, the paper is in particular at the beginning difficult to understand. Concepts such as the light node aren't clear. The term 'emaciated' used doesn't seem to have a clear technical meaning. On p.2, 'weak' devices are talked about. What are they (in relation to light nodes)?

Some other concepts such as tips or coordicide are used before they are properly explained. Again, this makes the paper hard to read.

Major criticisms apply to the state-of-the-art review and the novelty of the contribution.

The state-of-the-art is not adequately discussed. Only IOTA as a technology is introduced.

Also, the novelty in relation to related work is never sufficiently explained.

Furthermore, the selection of WLC is not sufficiently motivated. Following the authors' argumentation, the approach is suitable, but were there are options that could  have been considered?

The evaluation is not systematically introduced. What is needed is a clear specification of the evaluation objectives (is this only some sort of effectiveness or manage massive workloads?) and the strategy (e.g. simulation). Measurable indicators should be introduced.

The limitations should be better discussed. For example, the alpha setting to 0.5 (on p.9) should be discussed in a threats-to-validity section (together with other concerns that might impact on the validity of results) and/or should otherwise be listed as future work in the final conclusions.
Here technical device settings (resource-constrained IoT/edge devices) should be considered (cf. Section 4.2)

Some figures such as Fig. 5 are too small to be readable.

There are some English grammar issues. Often articles are missing. Proof-reading is recommended.

Reviewer 2 Report

Dear Authors

Your research is a good contribution, but you have to improve it.

  • In some sections, the text is piled up.
  • Better explain the methodology research used.
  • Use some type of graph or schema to better understand what you want to do. 
  • Improve the graphics, use another tool (R, Matlab, etc.), the quality of graphics is an expression of order and clarity. 
  • Explain the simulation used to obtain the results, this simulator has been validated? how? it is reliable?, it has been used in other investigations?, simulation parameters. 
  • Improve the conclusions.

Round 2

Reviewer 1 Report

My comments on the previous version have been satisfactorily addressed. I therefore recommend acceptance.

Reviewer 2 Report

Dear Authors

The research has been improved, now is much better.